Channel state information estimation for 5G wireless communication systems: recurrent neural networks approach

Essai Ali Mohamed Hassan mhessai@azhar.edu.eg 1
Taha Ibrahim B.M. 2
1 Department of Electrical Engineering, Faculty of Engineering, Al-Azhar University , Qena , Qena , Egypt
2 Department of Electrical Engineering, College of Engineering, Taif University , Taif , Saudi Arabia
Shang Yilun
Electronic publication date: 2021 Aug 26
Publication date: 2021
Volume: 7
Electronic Location ID: e682
Received 2021 May 3; Accepted 2021 Jul 29
Copyright: ©2021 Essai Ali and Taha
Copyright year: 2021
Copyright holder: Essai Ali and Taha
License: This is an open access article distributed under the terms of the Creative Commons Attribution License, which permits unrestricted use, distribution, reproduction and adaptation in any medium and for any purpose provided that it is properly attributed. For attribution, the original author(s), title, publication source (PeerJ Computer Science) and either DOI or URL of the article must be cited.
License URL: https://creativecommons.org/licenses/by/4.0/

Keywords: BiLSTM, Channel state information estimator, Deep learning neural networks, Loss functions

Funding: Taif University, Taif, Saudi Arabia, through the Taif University Researchers TURSP-2020/61 This work was supported by the Taif University, Taif, Saudi Arabia, through the Taif University Researchers Supporting Project, under Grant TURSP-2020/61. The funders had no role in study design, data collection and analysis, decision to publish, or preparation of the manuscript.

==============================
In this study, a deep learning bidirectional long short-term memory (BiLSTM) recurrent neural network-based channel state information estimator is proposed for 5G orthogonal frequency-division multiplexing systems. The proposed estimator is a pilot-dependent estimator and follows the online learning approach in the training phase and the offline approach in the practical implementation phase. The estimator does not deal with complete a priori certainty for channels’ statistics and attains superior performance in the presence of a limited number of pilots. A comparative study is conducted using three classification layers that use loss functions: mean absolute error, cross entropy function for kth mutually exclusive classes and sum of squared of the errors. The Adam, RMSProp, SGdm, and Adadelat optimisation algorithms are used to evaluate the performance of the proposed estimator using each classification layer. In terms of symbol error rate and accuracy metrics, the proposed estimator outperforms long short-term memory (LSTM) neural network-based channel state information, least squares and minimum mean square error estimators under different simulation conditions. The computational and training time complexities for deep learning BiLSTM- and LSTM-based estimators are provided. Given that the proposed estimator relies on the deep learning neural network approach, where it can analyse massive data, recognise statistical dependencies and characteristics, develop relationships between features and generalise the accrued knowledge for new datasets that it has not seen before, the approach is promising for any 5G and beyond communication system.

Introduction

5G wireless communication is the most active area of technology development and a rapidly growing branch of the wider field of communication systems. Wireless communication has made various possible services ranging from voice to multimedia.

The physical characteristics of the wireless communication channel and many unknown surrounding effects result in imperfections in the transmitted signals. For example, the transmitted signals experience reflections, diffractions, and scattering, which produce multipath signals with different delays, phase shift, attenuation, and distortion arriving at the receiving end; hence, they adversely affect the recovered signals (Oyerinde & Mneney, 2012).

A priori information on the physical characteristics of the channel provided by pilots is one of the significant factors that determine the efficiency of channel state information estimators (CSIEs). For instance, if not a priori information is available (no or insufficient pilots), channel estimation is useless; finding what you do not know is impossible. When complete information on the transmission channel is available, CSIEs are no longer needed. Thus, a priori uncertainty exists for communication channel statistics. However, the classical theory of detection, recognition, and estimation of signals deals with complete priory certainty for channel statistics, and it is an unreliable and unpractical assumption (Bogdanovich, Vostretsov & Electronics, 2009).

In the classic case, uncertainty is related to useful signals. In detection problems, the unknown is the fact of a signal existence. In recognition problems, the unknown is the type of signal being received at the current moment. In estimation problems, the unknown is the amplitude of the measured signal or one of its parameters. The rest of the components of the signal-noise environment in classical theory are regarded as a priori certain (known) as follows: the known is the statistical description of the noise, the known is the values of the unmeasured parameters of the signal and the known is the physical characteristics of the wireless communication channel. In such conditions, the classical theory allows the synthesis of optimal estimation algorithms, but the structure and quality coefficients of the algorithms depend on the values of the parameters of the signal-noise environment. If the values of the parameters describing the signal-noise environment are slightly different from the parameters for which the optimal algorithm is built, then the quality coefficients will become substantially poor, making the algorithm useless in several cases (Bogdanovich, Vostretsov & Electronics, 2009; O’Shea, Karra & Clancy, 2017). The most frequently used CSIEs are derived from signal and channel statistical models by employing techniques, such as maximum likelihood (ML), least squares (LS), and minimum mean squared error (MMSE) optimisation metrics (Kim, 2015).

One of the major concerns in the optimum performance of wireless communication systems is providing accurate channel state information (CSI) at the receiver end of the systems to detect the transmitted signal coherently. If CSI is unavailable at the receiver end, then the transmitted signal can only be demodulated and detected by a noncoherent technique, such as differential demodulation. However, using a noncoherent detection method occurs at the expense of a loss of signal-to-noise ratio of about 3–4 dB compared with using a coherent detection technique. To eliminate such losses, researchers have focused on the development of channel estimation techniques to provide perfect detection of transmitted information in wireless communication systems using the Orthogonal Frequency-Division Multiplexing (OFDM) modulation scheme (Oyerinde & Mneney, 2012).

The use of deep learning neural networks (DLNNs) is the state-of-the-art approach in the field of wireless communication. The amazing learning capabilities of DLNNs from training data sets and the tremendous progress of graphical processing units (GPUs), which are considered the most powerful tools for training DLNNs, have motivated its usage for different wireless communication issues, such as modulation recognition (Zhou, Liu & Gravelle, 2020; Karra, Kuzdeba & Petersen, 2017) and channel state estimation and detection (Essai Ali, 2021; Joo et al., 2019; Kang, Chun & Kim, 2020; Ma, Ye & Li, 2018; Ponnaluru & Penke, 2020; Yang et al., 2019; Ye, Li & Juang, 2018). According to Karra, Kuzdeba & Petersen (2017), Kim (2015), Oyerinde & Mneney (2012), Zhou, Liu & Gravelle (2020) and Ma, Ye & Li (2018), all proposed deep learning-based CSIEs have better performance compared with the examined traditional channel ones, such as LS and MMSE estimators.

Recently, numerous long short-term memory (LSTM)- and BiLSTM-based applications have been introduced for prognostic and health management (Zhao et al., 2020), artificial intelligence-based translation systems (Wu et al., 2016; Ong, 2017) and other areas. For channel state information estimation in 5G-OFDM wireless communication systems, many deep learning approaches, such as convolutional neural network (CNN), recurrent neural network (RNN) (e.g., LSTM and BiLSTM NNs) and hybrid (CNN and RNN) neural networks have been used (Essai Ali, 2021; Liao et al., 2019; Luo et al., 2018; Ponnaluru & Penke, 2020; Yang et al., 2019; Yang et al., 2019; Ye, Li & Juang, 2018).

In Liao et al. (2019), a deep learning-based CSIE was proposed by using CNN and BiLSTM-NN for the extraction of the feature vectors of the channel response and channel estimation, respectively. The aim was to improve the channel state information estimation performance at the downlink, which is caused by the fast time-varying and varying channel statistical characteristics in high-speed mobility scenarios. In Luo et al. (2018), an online-trained CSIE that is an integration of CNN and LSTM-NN was proposed. The authors also developed an offline–online training technique that applies to 5G wireless communication systems. In Ye, Li & Juang (2018), a joint channel estimator and detector that is based on feedforward DLNNs for frequency selective channel (OFDM) systems was introduced. The proposed algorithm was found to be superior to the traditional MMSE estimation method when unknown surrounding effects of communication systems are considered. In Yang et al. (2019), an online estimator was developed by adopting feedforward DLNNs for doubly selective channels. The proposed estimator was considered superior to the traditional LMMSE estimator in all investigated scenarios. In Ponnaluru & Penke (2020), a one-dimensional CNN (1D-CNN) deep learning estimator was proposed. Under various modulation scenarios and in terms of MSE and BER metrics, the authors compared the performance of the proposed estimator with that of feedforward neural networks (FFNN), MMSE and LS estimators. 1D-CNN outperformed LS, MMSE and FFNN estimators. In Essai Ali (2021), an online pilot-assisted estimator model for OFDM wireless communication systems was developed by using LSTM NN. The conducted comparative study showed the superior performance of the proposed estimator in comparison with LS and MMSE estimators under limited pilots and a priori uncertainty of channel statistics. Sarwar, Shah & Zafar (2020) used the genetic algorithm-optimised artificial neural network to build a CSIE. The proposed estimator was dedicated for space–time block-coding MIMO-OFDM communication systems. The proposed estimator outperformed LS and MMSE estimators in terms of BER at high SNRs, but it achieved approximately the same performance as LS and MMSE estimators at low SNRs. Senol, Tahir & Özmen (2021) proposed a CSIE for OFDM systems by using ANN under the condition of sparse multipath channels. The proposed estimator achieved a comparable SER performance as matching pursuit- and orthogonal matching pursuit-based estimators at a lower computational complexity than that of the examined estimators. Le Ha et al. (2021) proposed a CSIE that uses deep learning and LS estimator and utilizes the multiple-input multiple-output system for 5G-OFDM. The proposed estimator minimizes the MSE loss function between the LS-based channel estimation and the actual channel. The proposed estimator outperformed LS and LMMSE estimators in terms of BER and MSE metrics.

In this study, a BiLSTM DLNN-based CSIE for OFDM wireless communication systems is proposed and implemented. To the best of the authors’ knowledge, this work is the first to use the BiLSTM network as a CSIE without integration with CNN. The proposed estimator does not need any prior knowledge of the communication channel statistics and powerfully works at limited pilots (under the condition of less CSI). The proposed BiLSTM-based CSIE is a data-driven estimator, so it can analyse, recognise and understand the statistical characteristics of wireless channels suffering from many known interferences such as adjacent channel, inter symbol, inter user, inter cell, co-channel and electromagnetic interferences and unknown ones (Jeya et al., 2019; Sheikh, 2004). Although an impressively wide range of configurations can be found for almost every aspect of deep neural networks, the choice of loss function is underrepresented when addressing communication problems, and most studies and applications simply use the ‘log’ loss function (Janocha & Czarnecki, 2017). In this study two customed loss functions known as mean absolute error (MAE), and sum of squared errors (SSE) are proposed to obtain the most reliable and robust estimator under unknown channel statistical characteristics and limited pilot numbers.

The performance of the proposed BiLSTM-based estimator is compared with the performance of the most frequently used LS and MMSE channel state estimators. The obtained results show that the BiLSTM-based estimator attains a comparable performance as the MMSE estimator and outperforms LS and MMSE estimators at large and small numbers of pilots, respectively. In addition, the proposed estimator improves the transmission data rate of OFDM wireless communication systems because it exhibits optimal performance compared with the examined estimators at a small number of pilots.

The rest of this paper is organised as follows. The DLNN-based CSIE is presented in Section II. The standard OFDM system and the proposed deep learning BiLSTM NN-based CSIE are presented in Section III. The simulation results are given in Section IV. The conclusions and future work directions are provided in Section V.

DLNN-BASED CSIE

In this section, a deep learning BiLSTM NN for channel state information estimation is presented. The BiLSTM network is another version of LSTM neural networks, which are recurrent neural networks (RNN) that can learn the long-term dependencies between the time steps of input data (Hochreiter & Schmidhuber, 1997; Luo et al., 2018; Zhao et al., 2020).

The BiLSTM architecture mainly consists of two separate LSTM-NNs and has two propagation directions (forward and backward). The LSTM NN structure consists of input, output and forget gates and a memory cell. The forget and input gates enable the LSTM NN to effectively store long-term memory. Figure 1 shows the main construction of the LSTM cell (Hochreiter & Schmidhuber, 1997). The forget gate enables LSTM NN to remove the undesired information by currently used input xt and cell output ht of the last process. The input gate finds the information that will be used with the previous LSTM cell state ct−1 to obtain a new cell state ct based on the current cell input xt and the previous cell output ht−1. Using the forget and input gates, LSTM can decide which information is abandoned and which is retained.

Figure 1 Long short-term memory (LSTM) cell.

The output gate finds current cell output ht by using the previous cell output ht−1 at current cell state ct and input xt. The mathematical model of the LSTMNN structure can be described through Eqs. (1) –(6). (1) it=σgwixt+Riht−1+bi

(2) ft=σgwfxt+Rfht−1+bf

(3) gt=σcwgxt+Rght−1+bg

(4) ot=σgwoxt+Roht−1+bo

(5) ct=ft⨀ct−1+it⨀gt

(6) ht=ot⨀σcct

where i, f, g, o,  σc, σg and ⨀ denote the input gate, forget gate, cell candidate, output gate, state activation function (hyperbolic tangent function (tanh), gate activation function (sigmoid function) and Hadamard product (element-wise multiplication of vectors), respectively. W = [wiwfwgwo]T, R = [RiRfRgRo]T and b = [bibfbgbo]T are input weights, recurrent weights and bias, respectively.

LSTM DNN, only analyses the impact of the previous sequence in the present, disregarding information later on and failing to reach optimal performance. On the other hand BiLSTM connects the LSTM unit’s output bidirectionally (forward and backward propagation directions) and capture bidirectional signals dependencies, increasing the overall model’s performance.

The forward and backward propagation directions of BiLSTM are transmitted at the same time to the output unit. Therefore, old and future information can be captured, as shown in Fig. 2. At any time t, the input is fed to forward LSTM and backward LSTM networks. The final output of BiLSTM-NN can be expressed as follows: (7) ht=ht →⨀ht →,

where h→t and h→t are forward and backward outputs of BiLSTM-NN, respectively. The operation of BiLSTM in the proposed estimator can be described briefly by the following algorithm:

Figure 2 BiLSTM-NN architecture.

Input: sequence represents transmitted signal (original signal + channel model)

Output: Prediction matrix of the extracted features of the input sequence

Step 1: The forward LSTM layer receives the transmitted signal vectors from X.

for i ∈length (X) do

send Xi to BiLSTM Layer

end for

Step 2: Eqs. (1)–(6) are used to update the state of the LSTM cell.

Step 3: The backward LSTM layer receives the signal vectors from X, and the two previous steps are repeated.

Step 4: A hidden state sequence vector is created by splicing the forward and backward sequences of hidden layers.

Step 5: A hidden state sequence vector is sent into a full connection layer and the prediction matrix is obtained

Step 6: Return the prediction matrix.

To build the DL BiLSTM NN-based CSIE, an array is created with the following five layers: sequence input, BiLSTM, fully connected, softmax and output classification. The input size was set to 256. The BiLSTM layer consists of 30 hidden units and shows the sequence’s last element. Four classes are specified by considering the size 4 fully connected (FC) layer, followed by a softmax layer and ended by a classification layer. Figure 3 illustrates the structure of the proposed estimator (Essai Ali, 2021; Ye, Li & Juang, 2018).

Figure 3 Structure of the DL BiLSTM NN for the BiLSTM estimator.

As the proposed BiLSTM-based CSIE is built, the weights and biases of the proposed estimator are optimised (tuned) using the desired optimisation algorithm. The optimisation algorithm trains the proposed estimator by using one of three loss functions, namely, cross entropy function for kth mutually exclusive classes (crossentropyex), mean absolute error (MAE), and sum of squared errors (SSE). The loss function estimates the loss between the expected and actual outcome. During the learning process, optimisation algorithms try to minimise the available loss function to the desired error goal by optimising the DLNN weights and biases iteratively at each training epoch. Figure 4 illustrates the training processes of the proposed estimator. Selecting a loss function is one of the essential and challenging tasks in deep learning. Also, investigating the efficiency of the training process using different optimization algorithms such as Adaptive Moment Estimation (Adam), Root Mean Square Propagation (RMSProp), Stochastic Gradient Descent with momentum (SGdm) (Dogo et al., 2018), and an adaptive learning rate method (Adadelta) (Zeiler, 2012). The proposed estimator is trained using above-mentioned three different loss functions and optimization algorithms to obtain the most optimal BiLSTM-based estimator for wireless communication systems with low prior information (limited pilots) for signal-noise environments.

Figure 4 Offline training of the BiLSTM-NN-based CSI estimator.

DL BiLSTM NN-Based CSIE for 5G–OFDM wireless communication systems

The standard OFDM wireless communication system and an offline DL of the proposed CSIE are presented in the following subsections.

OFDM System Model

In accordance with Essai Ali (2021) and Ye, Li & Juang (2018), Fig. 5 clearly illustrates the structure of the traditional OFDM communication system. On the transmitter side, a serial-to-parallel (S/P) converter is used to convert the transmitted symbols with pilot signals into parallel data streams. Then, inverse discrete Fourier transform (IDFT) is applied to convert the signal into the time domain. A cyclic prefix (CP) must be added to alleviate the effects of inter-symbol interference. The length of the CP must be longer than the maximum spreading delay of the channel.

Figure 5 Conventional OFDM system.

The multipath channel of a sample space defined by complex random variables hnn=0N−1 is considered. Then, the received signal can be evaluated as follows: (8) yn=xn⊕hn+wn,

where ⊕x(n) is the input signal, ⊕ is circular convolution, w(n) is additive white Gaussian noise (AWGN) and y(n) is the output signal.

The received signal in the frequency domain can be defined as (9) Yk=XkHk+Wk,

where the discrete Fourier transformations (DFT) of x(n), h(n), y(n) and w(n) are X(k), H(k), Y(k) and W(k), respectively. These discrete Fourier transformations are estimated after removing CP.

The OFDM frame includes the pilot symbols of the 1st OFDM block and the transmitted data of the next OFDM blocks. The channel can be considered stationary during a certain frame, but it can change between different frames. The proposed DL BiLSTM NN-based CSIE receives the arrived data at its input terminal and extracts the transmitted data at its output terminal (Essai Ali, 2021; Ye, Li & Juang, 2018).

OFFLINE DL OF THE DL BILSTM NN-BASED CSIE

DLNN utilisation is the state-of-the-art approach in the field of wireless communication, but DLNNs have high computational complexity and long training time. GPUs are the most powerful tools used for training DLNNs (Sharma, Vinutha & Moharir, 2016). Training should be done offline due to the long training time of the proposed CSIE and the large number of BILSTM-NN’s parameters, such as biases and weights, that should be tuned during training. The trained CSIE is then used in online implementation to extract the transmitted data (Ye, Li & Juang, 2018; Essai Ali, 2021).

In offline training, the learning dataset is randomly generated for one subcarrier. The transmitting end sends OFDM frames to the receiving end through the adopted (simulated) channel, where each frame consists of single OFDM pilot symbol and a single OFDM data symbol. The received OFDM signal is extracted based on OFDM frames that are subjected to different channel imperfections.

All classical estimators rely highly on tractable mathematical channel models, which are assumed to be linear, stationary and follow Gaussian statistics. However, practical wireless communication systems have other imperfections and unknown surrounding effects that cannot be tackled well by accurate channel models; therefore, researchers have developed various channel models that effectively characterise practical channel statistics. By using these channel models, reliable and practical training datasets can be obtained by modelling (Bogdanovich, Vostretsov & Electronics, 2009; Essai Ali, 2021; 2019).

In this study, the 3GPP TR38.901-5G channel model developed by (2019) is used to simulate the behaviour of a practical wireless channel that can degrade the performance of CSIEs and hence, the overall communication system’s performance.

The proposed estimator is trained via the algorithm, which updates the weights and biases by minimising a specific loss function. Simply, a loss function is defined as the difference between the estimator’s responses and the original transmitted data. The loss function can be represented by several functions. MATLAB/neural network toolbox allows the user to choose a loss function amongst its available list that contains crossentropyex, MSE, sigmoid and softmax. In this study, another two custom loss functions (MAE and SSE) are created. The performance of the proposed estimator when using three loss functions (i.e., MAE, crossentropyex and SSE) is investigated. The loss functions can be expressed as follows: (10) crossentropyex=−∑i=1N ∑j=1cXijklogX ˆijk,

(11) MAE=∑i=1N ∑j=1cXijk−X ˆijkN,

(12) SSE= ∑i=1N ∑j=1cXijk−X ˆijk2,

where N is the sample number, c is the class number, Xij is the ith transmitted data sample for the jth class and X ˆij X ˆij is the DL BiLSTM-based CSIE response for sample i ifor class j.

Figure 4 illustrates the offline training processes to obtain a learned CSIE based on BiLSTM-NN.

Simulation Results

Studying the performance of the proposed, LS and MMSE estimators by using different pilots and loss functions

Several simulation experiments are performed to evaluate the performance of the proposed estimator. In terms of symbol error rate (SER) performance analysis, the SER performance of the proposed estimator under various SNRs is compared with that of the LSTM NN-based CSIE (Essai Ali, 2021), the well-known LS estimator and the MMSE estimator, which is an optimal estimator but requires channel statistical information. A priori uncertainty of the used channel model statistics is assumed and considered for all conducted experiments.

Moreover, the Adam optimisation algorithm is used to train the proposed estimator whilst using different loss functions to obtain the most robust version of the proposed CSIE. The proposed model is implemented in 2019b MATLAB/software.

Table 1 lists the parameters of BiLSTM-NN and LSTM-NN architectures and their related training options. These parameters are identified by a trial-and-error approach. Table 2 lists the parameters of the OFDM system model and the channel model.

Table 1 BiLSTM- and LSTM-NN structure parameters and training process options.

Parameter	Value	
Input Size	256	
BiLSTM Layer Size	30 hidden neurons	
LSTM Layer Size	30 hidden neurons	
FC Layer Size	4	
Loss Functions	Crossentropyex, MAE, SSE	
Mini Batch Size	1000	
Epochs Number	1000	
Learning Algorithm	Adam	
Training Data Size	8000 - OFDM frame	
Validation Data Size	2000 - OFDM frame	
Test Data Size	10000 - OFDM frame	

Table 2 OFDM system and channel parameters.

Parameter	Value	
Modulation Mode	QPSK	
Carrier Frequency	2.6 GHz	
Paths Number	24	
CP Length	16	
Subcarrier Number	64	
Pilot Number	64, 8 and 4	

The examined estimators’ performance is evaluated at different pilot numbers of 4, 8 and 64 as well as crossentropyex, MAE and SSE loss functions. The Adam optimisation algorithm is used for all simulation experiments.

With a sufficiently large number of pilots (64) and the use of the crossentropyex loss function, the proposed BiLSTMcrossentropyex estimator outperforms LSTMcrossentropyex, LS and MMSE estimators over the entire SNR range, as shown in Fig. 6. At the use of the MAE loss function, the BiLSTMMAE estimator outperforms the LS estimator over the SNR range [0–18 dB], but LSTMMAE outperforms it over the SNR range [0–14 dB]. In addition, the BiLSTMMAE and LSTMMAE estimators are at par with the MMSE estimator over the SNR ranges [0–10 dB] and [0–4 dB], respectively. Beyond these SNR ranges, the MMSE estimator outperforms BiLSTMMAE and LSTM MAE estimators. BiLSTMMAE outperforms LSTMMAE starting from 0 dB to 20 dB.

Figure 6 SER comparison of LS, MMSE, BiLSTM and LSTM estimators using 64 pilots, the Adam learning algorithm and crossentropyex, MAE and SSE loss functions.

At the use of the SSE loss function, Fig. 6 shows that the BiLSTMSSE and LSTMSSE estimators achieve approximately the same performance as the MMSE estimator over a low SNR range [0–6 dB]. MMSE outperforms the BiLSTMSSE and LSTMSSE estimators starting from 8 dB, and the LS estimator outperforms BiLSTMSSE starting from 16 dB and LSTMSSE starting from 14 dB. BiLSTMSSE outperforms LSTMSSE starting from 10 dB to 20 dB. LS provides poor performance compared with MMSE because it does not use prior information about channel statistics in the estimation process. MMSE exhibits superior performance, especially with sufficient pilot numbers, because it uses second-order channel statistics. Concisely, MMSE and the proposed BiLSTMcrossentropyex attain close SER performance with respect to all SNRs. Furthermore, at low SNR (0–6 dB), BiLSTM(crossentropyex, MAE, and SSE), LSTM(crossentropyex, MAE, and SSE) and MMSE attain approximately the same performance.

Figure 7 present the performance comparison of LS, MMSE, BiLSTM and LSTM-based estimators using the Adam optimisation algorithm and the different (crossentropyex, MAE and SSE) loss functions at 8 pilots. Figure 7 shows that the proposed BiLSTM(crossentropyex, or MAE or SSE) estimators outperform the LSTM(crossentropyex, or MAE or SSE) estimators and the traditional estimators over the examined SNR range. At a low SNR (0–7 dB), the proposed BiLSTM(crossentropyex, or MAE or SSE) estimators exhibit semi-identical performance. Furthermore, the proposed BiLSTMSSE estimator trained by minimising the SSE loss function outperforms the BiLSTMcrossentropyex estimator trained by minimising the crossentropyex loss function starting from 0 dB; also it outperforms BiLSTMMAE, which is trained by minimising the MAE loss function starting from 14 dB. Concisely at 8 pilots BiLSTMSSE estimator achieved the most minimum SER.

Figure 7 SER performance comparison of LS, MMSE, BiLSTM, and LSTM estimators using 8 pilots, the Adam learning algorithm and crossentropyex, MAE and SSE loss functions.

Figure 8 show the performance comparison of the LS, MMSE, BiLSTM(crossentropyex, or MAE or SSE) and LSTM(crossentropyex, or MAE or SSE) estimators at four pilots. Figure 8 shows the superiority of the proposed BiLSTM(crossentropyex, or MAE or SSE) estimators in comparison with the traditional estimators, which have lost their workability starting from 0 dB. It also shows the superiority of the proposed estimator BiLSTM(MAE or SSE) over LSTM(MAE or SSE). LSTM(crossentropyex) exhibits a competitive performance as BiLSTM(crossentropyex) starting from 0 dB to 12 dB, and LSTM(crossentropyex) outperforms BiLSTM(crossentropyex) starting from 14 dB. At very low SNRs (0–3 dB), the proposed BiLSTM(crossentropyex, or MAE or SSE) estimators have the same performance. The proposed BiLSTMSSE estimator outperforms the BiLSTMcrossentropyex estimator starting from 4 dB, and it exhibits an identical performance as the BiLSTMMAE estimator until 14 dB and outperforms it in the rest of the SNR examination range.

Figure 8 SER performance comparison of LS, MMSE, BiLSTM, and LSTM estimators using 4 pilots, the Adam learning algorithm and crossentropyex, MAE and SSE loss functions.

Figures 6–8 emphasise the robustness of the BiLSTM-based estimators against the limited number of pilots, low SNR, and under the condition of a priori uncertainty of channel statistics. They demonstrate the importance of testing various loss functions in the deep learning process to obtain the most optimal architecture of any proposed estimator.

Figure 9 indicates that the proposed BiLSTMcrossentropyex, BiLSTMSSE and BiLSTMSSE estimators have close SER performance at 64, eight and four pilots, respectively. The performance of BiLSTMSSE at eight pilots coincides with the performance of BiLSTMcrossentropyex at 64 pilots. Therefore, using the proposed estimators with few pilots is recommended for 5G OFDM wireless communication systems to attain a significant improvement in their transmission data rate. Given that the proposed estimator adopts a training data set-driven approach, it is robust to a priori uncertainty for channel statistics.

Figure 9 SER performance comparison of the best DL BiLSTM-based CSIEs using various pilots and loss functions.

Figure 10 Loss curves comparison of BiLSTM- and LSTM- based estimators using 64 pilots, the Adam learning algorithm and crossentropyex, MAE and SSE loss functions.

Figure 11 Loss curves comparison of BiLSTM- and LSTM-based estimators using eight pilots, the Adam learning algorithm and crossentropyex, MAE and SSE loss functions.

Figure 12 Loss curves comparison of BiLSTM- and LSTM-based estimators using four pilots, the Adam learning algorithm and crossentropyex, MAE and SSE loss functions.

Loss curves

The quality of the DLNNs’ training process can be monitored efficiently by exploring the training loss curves. These loss curves provide information on how the training process goes, and the user can decide whether to let the training process continue or stop.

Figures 10–12 show the loss curves of the DLNN-based estimators (BiLSTM and LSTM) at pilot numbers = 64, eight and four and with the three examined loss functions (crossentropyex, MAE and SSE). The curves emphasise and verify the obtained results in Figs. 6, 7, and 8. For example, the sub-curves in Fig. 10 for BiLSTMcrossentropyex and LSTMcrossentropyex estimators emphasise their superiority over the other estimators. This superiority can be seen clearly from Fig. 6. Moreover, the training loss curves in Figs. 11 and 12 emphasise the obtained SER performance in Figs. 7 and 8, respectively, of each examined DLNN-based CSIE. For more details, good zooming, and analysis of the presented loss curves, they can be downloaded from this link (shorturl.at/lqxGQ).

Accuracy calculation

The accuracy of the proposed and other examined estimators is a measure of how the estimators recover transmitted data correctly. Accuracy can be defined as the number of correctly received symbols divided by the total number of transmitted symbols. The proposed estimator is trained in different conditions as indicated in the previous subsection, and we wish to investigate how well it performs in a new data set. Tables 3, 4 and 5 present the obtained accuracies for all examined estimators under all simulation conditions.

Table 3 Accuracy comparison of the examined estimators using 64 pilots.

64 pilots	
	BiLSTM	LSTM	MMSE	LS	
Crossentropyex	100	99.99	100	99.94	
SSE	99.23	97.88	100	99.96	
MAE	99.87	99.52	100	99.97	

Table 4 Accuracy comparison of the examined estimators using eight pilots.

8 pilots	
	BiLSTM	LSTM	MMSE	LS	
Crossentropyex	99.84	99.53	91.34	91.62	
SSE	100	99.95	91.60	91.49	
MAE	100	99.94	91.53	91.50	

Table 5 Accuracy comparison of the examined estimators using four pilots.

4 pilots	
	BiLSTM	LSTM	MMSE	LS	
Crossentropyex	98.61	97.94	0.24	0.02	
SSE	100	99.28	0.24	0.09	
MAE	99.97	99.05	0.26	0.04	

As illustrated in Tables 3 to 5, the proposed BiLSTM-based estimator attains accuracies from 98.61 to 100 under different pilots and loss functions. The other examined DL LSTM-based estimator has accuracies from 97.88 to 99.99 under the same examination conditions. The achieved accuracies indicate that the proposed estimator has robustly learned and emphasises the obtained SER performance in Fig. 9. The obtained results of MMSE and LS in Tables 1, 2 and 3 emphasise the presented SER performance in Figs. 6, 7 and 8, respectively, and show that as the pilot number decreases, the accuracy of the conventional estimators dramatically decreases.

The proposed BiLSTM- and LSTM-based estimators rely on DLNN approaches, where they can analyse huge data sets that may be collected from any plant, recognise the statistical dependencies and characteristics, devise the relationships between features and generalise the accrued knowledge for new data sets that they have not seen before. Thus, they are applicable to any 5G and beyond communication system.

Impact of using different optimization algorithms on the proposed estmator performance

DL procedures benefit greatly from optimization methods. DNN training can be thought of as an optimisation issue that aims to discover a global optimum by applying gradient descent methods to obtain a robust training, and hence reliable prediction or classification models. Choosing the best optimization method for a particular scientific topic is a difficult task. Using the wrong optimization strategy during training can cause the DN to stay at the local minimum, which results in no training progress (Dogo et al., 2018). As a result, examination is required to evaluate the performance of various optimisers to get the optimal CSIE.

This section provides performance comparison experiments using RMSProp, SGdm, and Adadelta optimisation algorithms (Soydaner & Intelligence, 2020)for training the proposed BiLSTM-based CSIE at using 8-pilots, as illustrated in Fig. 13. Table 6 arranges the proposed BiLSTM CSIE estimators using different optimisation algorithms and loss functions from the highest performance to the lowest and their related accuracies.

Figure 13 Performance comparison of BiLSTM-based estimator using eight pilots, the RMSProp, SGdm, and Adadelta optimisation algorithms and crossentropyex, MAE and SSE loss functions.

Table 6 Performance comparison of different optimisation algorithms and its related accuracies.

Order	Optimisation algorithm Loss function	Accuracy	
First	AdadeltaSSE	100%	
Second	Adadeltacrossentropyex	99.99%	
Third	AdadeltaMAE	99.98%	
Fourth	RMSPropcrossentropyex	99.90%	
Fifth	RMSPropMAE	99.84%	
Sixth	RMSPropSSE	99.74%	
Seventh	SGdmMAE	98.76%	
Eighth	SGdmcrossentropyex	98.53%	
Ninth	SGdmSSE	97.46%	

It is clear from Fig. 13 and Table 6 that the trained BiLSTM-based CSIE using Adadelta optimisation algorithm and SSE loss function achieves the best SER performance and provides the highest accuracy with 100%. On the other hand, the same estimator achieves the lowest SER performance and provides accuracy with 97.46% using SGdm optimization algorithm and SSE loss function. This, in turn, shows the importance of studying the training process efficiency using different optimization algorithms in the case of using a specific loss function.

Conclusions and Future Work

The proposed DL-BiLSTM-based CSIE is an online pilot-assisted estimator. It is robust against a limited number of pilots and exhibits superior performance compared with conventional estimators; it is also robust under the conditions of a priori uncertainty of communication channel statistics (non-Gaussian/stationary statistical channels) and demonstrates superior performance compared with conventional estimators and DL LSTM NN-based CSIEs.

Two customized classification layers using the loss functions (MAE and SSE) are introduced. The proposed CSIE exhibits a consistent performance at large and small pilot numbers and superior performance at low SNRs, especially at limited pilots, compared with conventional estimators. It also achieves the highest accuracy amongst all examined estimators at 64, eight, and four pilots for all the used loss functions.

The proposed BiLSTM- and LSTM-based estimators have high prediction accuracies of 98.61% to 100% and 97.88% to 99.99%, respectively, when using crossentropyex, MAE, and SSE loss functions for 64, eight, and four pilots. The proposed BiLSTM using (Adam, and crossentroyex), BiLSTM using (Adam, MAE, and SSE; and Adadelta, and SSE), and BiLSTM using (Adam, and SSE), achieve the best SER performance and provide accuracies with 100% at 64, eight, and four pilots respectively. The proposed estimator is promising for 5G and beyond wireless communication systems.

For future work, authors suggest the following research plans:

1. Investigating the proposed estimator’s performance and accuracy by using different cyclic prefix lengths and types.

2. Developing robust loss functions by using robust statistics estimators, such as Tukey, Cauchy, Huber and Welsh.

3. Investigating the performance of CNN-, gated recurrent unit (GRU)- and simple recurrent unit (SRU)-based CSIEs whilst using crossentropyex, MAE and SSE loss functions and for 64, eight, and four pilots.

Supplemental Information

Supplemental Information 1 Testing the trained Net and calculating accuracies

Click here for additional data file.

Supplemental Information 2 Read me

Click here for additional data file.

Supplemental Information 3 3GPP channel model

Click here for additional data file.

Supplemental Information 4 The channel matrix generated using the 3GPP TR38.901 channel model of the writer’s own implementation, h parameter

Click here for additional data file.

Supplemental Information 5 The channel matrix generated using the 3GPP TR38.901 channel model of the writer’s own implementation, idxSC parameter

Click here for additional data file.

Supplemental Information 6 This function is to transform the received OFDM packets to feature vectors for training and collect the corresponding labels

Code

Click here for additional data file.

Supplemental Information 7 Custom classification layer with mean-absolute-error loss

Click here for additional data file.

Supplemental Information 8 This function is to model the transmission and reception process in OFDM systems

Click here for additional data file.

Supplemental Information 9 The channel matrix generated using the 3GPP TR38.901 channel model of the writer’s own implementation, which is saved and loaded

Click here for additional data file.

Supplemental Information 10 3GPP channel model

Click here for additional data file.

Supplemental Information 11 Custom classification layer with sum of squares error loss

Click here for additional data file.

Supplemental Information 12 This script is to set up parameters for training the BiLSTM or LSTM deep neural network for the selected subcarrier based on the training data

Click here for additional data file.

Supplemental Information 13 This script is created to generate training and validation data for the deep learning model

Click here for additional data file.

Additional Information and Declarations

Competing Interests

Author Contributions

Data Availability

The authors declare there are no competing interests.

Mohamed Hassan Essai Ali conceived and designed the experiments, performed the experiments, analyzed the data, performed the computation work, prepared figures and/or tables, authored or reviewed drafts of the paper, and approved the final draft.

Ibrahim B.M. Taha conceived and designed the experiments, performed the experiments, analyzed the data, prepared figures and/or tables, authored or reviewed drafts of the paper, and approved the final draft.

The following information was supplied regarding data availability:

Matlab code is available in the Supplemental Files.

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
