# Peer review of "Channel state information estimation for 5G wireless communication systems: recurrent neural networks approach"

_PeerJ Computer Science, doi:10.7717/peerj-cs.682_

## Round 0.1 · original submission · Major Revisions

Three reviews have been received for this paper. Reviewers identified some good merits of the paper but some weak points have also been raised. The methodology and experiments should be clarified. Please provide a detailed point to point response to the reviewers.

·

Basic reporting

The paper is well written. The results are clear.
Only proofreading is required.

Experimental design

well presented.

Validity of the findings

well presented.

Additional comments

Please proof read the paper.

Reviewer 2 ·

Basic reporting

Some proper references are required for lines 44-48; for lines 54-56; for lines 91-94.

OFDM is just given by an abbreviation; an open form is required in line 80.

In line 144, what are the known interferences that occur on this kind of system? Any reference on this? For this claim, it seems an inexplicit expression.

You should remove the text in line 153.

Figure 1 needs a cite.

Figure 2 has low quality.

Line 207 ….. .On -> a space after full stop

For Figure 4, details should be increased. Legend of the blocks, etc., are required. In the text, serial-to-parallel (probably) refers to S/P, but a more technical perspective of all figures up to that point is a must.

CP is probably a cyclic prefix, but it is not indicated in the text properly. (Line 220 & 209)
The most crucial lines in the paper are: 188-191 & 223-224. They are telling the contribution proposed by the authors; however, they all can be expanded by adding some flow charts, further explanation, etc.

The attached code requires a readme to show the procedures of the .m file hierarchy. For instance, I keep taking an "abstract class instantiation" error for sseClassificationLayer.m (even if I have corrected backwardLoss(), there occur some other problems throwing an error. My MATLAB is R2018b).

Please re-check some typos in your attached code,
like: "%plotting the loss and accuracy curves 'sparately' " (TrainDNN.m)
or : "% in the MATLAB example of 'seqeunce' classification using LSTM network."
"% is represented by a feature vector that follows the similar data 'struture' "
etc.

Experimental design

The repeatedly highlighted Adam optimization has no big impact on the overall contribution. What about RMSProp, AdaGrad, etc.?

I did not get the point of using BiLSTM instead of LSTM. Why/How your system requires a bidirectional movement on the timeline?

What are the data details? Data size & training, validation, testing portions?
(Even if I can get the information through supplementary material, the paper itself should present all this information properly.)

What about further network details like dropout, batch normalization?

Validity of the findings

For Figure 7, why is there a transition for the best performing techniques: BiLSTM versus LSTM? Is there any explanation for this? Or is this just experimentally obtained observation?

Results based on Figures 6-18 should be summarized in a table like indicating the effect of "the applied method", "pilot numbers", and "dB range". A performance comparison heat-map table can be an idea.

The complexity section does not give critical feedback.

Loss curves have the zoom problem; the only critical regions of each network can be plotted side-by-side. Since for some certain epochs, some of the network architectures do not indicate significant learning.

Conclusions and future work sections can be merged. Plus, item-based representation is a strange way for the overall section. There may be some significant contributions as items, but not the overall section. (To the best of my knowledge, PeerJ does not demand a listing style like this.)

Additional comments

For the introduction part;
While reading the paper, the transition between the communication systems and the deep neural network architectures seems quite abstract. This may be because the current structure of the introduction section is like a literature review. I suggest that readers either technically explain the relation between BiLSTM and the CSIE or change the current structure of the introduction to give just the definition of the problem, a slight literature summary, and the main points of their proposal. I highly recommend the latter since the introduction is like mixing several different information. For instance, why should a reader get information about the “loss function” before thoroughly knowing the problem definition?

Line 192-197: Did you decide on this structure based on any preliminary experiments? Or some of the previous efforts (even though you claim to be the first on this structure; maybe you inspired by any LSTM-based communication system.) Why a structure like that? This should be technically explained in this paragraph.

Figure 3 should have been more detailed such as drawing with more information on layer size, activation function, connections, etc. Figure 3 (b) does not tell scientific information.

Information in line 250 about MATLAB seems strange since the reader cannot get the point whether authors mention a neural net toolbox, or Simulink, or plain code – built-in methods, etc.

The utilization of different loss functions, Adam optimization, etc., cannot be considered as the main contributions of the study. If authors would have widely present training & validation procedures, it may be possible to mark the paper as a well-contributing research paper. For this current version, it seems like “investigating the NN performance with try and error methods”.

For the further efforts of this paper, I strongly suggest including the training parameters and a wide-angle figure of the proposed network architecture, including every single detail from input to the output (such as which signals are exactly sourced, network hyperparameters, connections, dropout, batch normalization, etc.). Besides, the way the simulations are handled is far away from being a vivid picture. A summary of the plots into the table (heat-map or ticking for performances like ✓ ✓✓ ✓✓✓ ✓✓✓✓ , etc. ) can be an idea to present "the applied method", "pilot numbers", and "dB range" performance.

Reviewer 3 ·

Basic reporting

no comment

Experimental design

no comment

Validity of the findings

no comment

Additional comments

The manuscript is providing the practical use of BiLSTM networks for the 5G systems which is interesting. However, some additional improvements are required.

1) The written text must be improved. For example in the introduction parts the paragraph divisions are not suitable and must be organized well.

2) The conclusion and future parts must started with some sentences and then start using i ii, etc.

3) The quality of the figures must be improved well.

4) Please put the comparison table and compare the results with other reported high impact factor papers.

5) Please add some up-to-date journals in the reference list and add them in the introduction part. Then please discuss about the drawbacks with that references and clarify your contributions.

---

## Round 0.2 · Major Revisions

The reviewers have provided useful comments on the merits and drawbacks of the paper. There are still some places that need to be rectified, especially for Reviewer 2. Please give detailed one to one responses to the reviewers.

Reviewer 2 ·

Basic reporting

My previous demand:
"The most crucial lines in the paper are: 188-191 & 223-224. They are telling the contribution proposed by the authors; however, they all can be expanded by adding some flow charts, further explanation, etc."

Author response:
"Authors think that main contributions are declared enough before, through, and after these lines."

Based on my previous review, I had indicated the main contribution explanatory paragraphs and demanded: "however, they all can be expanded by adding some flow charts, further explanation, etc." Visually, it is better to add a "motivation of the study of things." This does not mean "Authors think that main contributions are declared enough before, though, and after these lines." as you have claimed.
* * *
My previous question:
The complexity section does not give critical feedback.

Author response:
This section represents a complementary study for this type of work. The section presents much valuable information, such as training time and the feed-forward pass and feedback pass operations complexity for BiLSTM, and LSTM-based estimators.

You indicate O(w) -> O(2w) when it becomes “Bidirectional”. What does this bring to the reader? One can guess that if the direction of data processing in a network increases, the timing will also; plus, this is not a new finding. The formulation you utilize; I am not familiar with, please see:
- http://cse.iitkgp.ac.in/~psraja/FNNs%20,RNNs%20,LSTM%20and%20BLSTM.pdf
- https://arxiv.org/pdf/2103.08212.pdf

Besides, Table 6 is filled with some ratios? What is ":" operator for?
If you could find a proper citation for your highlight on the complexity, giving a cite is welcomed.
* * *
My previous question:
Line 192-197: Did you decide on this structure based on any preliminary experiments? Or some of the previous efforts (even though you claim to be the first on this structure; maybe you inspired by any LSTM-based communication system.) Why a structure like that? This should be technically explained in this paragraph.

Author response:
Yes, I decided on this structure based on preliminary experiments and some of the previous efforts. Kindly refer to Ref. [12]. Kindly refer to "Revised manuscript" file.

Then, why not clearly include this inspiration in the place where the first time the network architecture is mentioned (If I am not wrong, Ref. 12 is Liao et al. is only occurring in the introduction)

Experimental design

My previous question:
The repeatedly highlighted Adam optimization has no big impact on the overall contribution. What about RMSProp, AdaGrad, etc.?

Author response:
The use of both RMSProp, AdaGrad, and other optimization algorithms such Adadelta, Adagrad, AMSgrad, AdaMax and Nadam is suggested in the future work section. Kindly refer to "Future work".

I do not agree with the authors to save some information for future work; at the end of the day, these issues are not mitigated and the reader still does not know anything about other optimizers. It is better to put a comparison results (if saying repeatedly about Adam’s optimizer), or do not highlight too much of Adam's optimization. Nevertheless, for my overall expectation, as you will see at the end, the comparison result is more expected.

Validity of the findings

My one other previous demand is still not clear:
I did not get the point of using BiLSTM instead of LSTM. Why/How your system requires a bidirectional movement on the timeline?

Author response:
All obtained results demonstrate the superior performance of BiLSTM-based estimators compared with conventional estimators and LSTM-based estimators, so the study recommends using BiLSTM-based estimators, especially at low signal-to-noise ratios.

Still, it seems to me “it has been tried and the results are like”. I am asking more mathematical perspective, what are the signal properties of LSTM & BiLSTM; why a bidirectional movement is required on a communication system as indicated. Are there any other advantages at least to use BiLSTM? Each and every day a new NN algorithm is being proposed; next time one other researcher can try enhanced BiLSTM, etc., but this does not seem to be a contribution; at least in your paper, more than the experimental results, one can infer why and how to construct a communication system based on BiLSTM. (For instance, at least you may include an Appendix (diagram, UML, or flowchart) by showing your code structure; thereby giving the related math becomes visually possible)

Additional comments

For the author response:
“Since the behavior of neural networks relies on different factors such as the network structure, the learning algorithm, loss functions, the activation functions used in each node, etc.,”

I agree for a research paper to include preliminary insights for the NN & Comm. Syst. collaborations; however, as you have given with your previous paper as you have shared, a thorough inspection on all these parameters is required. There are “different factors” as you have said; however, I cannot catch the main summary of all these factors as I had demanded with my sentences starting “For the further efforts of this paper, I strongly suggest”. You have taken it as a future work; however, my main aim is for this current paper.

Therefore, I still believe that a complete summary of the performance related to the indicated factors should be given. I believe this is the only way to increase the contribution of your paper. You may present a summary of all your experiments (by even increasing) showing the effects of all different loss functions, optimizers, learning rate, network size, etc. At least, you may choose one single architecture like BiLSTM 8, and make a table of all NN dynamics.

By the way, thank you very much for the heat map tables, now, I believe that the related results are clearer.

Reviewer 3 ·

Basic reporting

no comment

Experimental design

no comment

Validity of the findings

no comment

Additional comments

The manuscript is well written and all the required editions are done.

---

## Round 0.3 · accepted · Accept

The final reviewer has agreed to accept the paper. Congratulations!

Reviewer 2 ·

Basic reporting

All issues related to the previous round have been resolved by the authors.

Experimental design

All issues related to the previous round have been resolved by the authors.

Validity of the findings

All issues related to the previous round have been resolved by the authors.